# Augmented Aircraft Performance with the Use of Morphing Technology for a Turboprop Regional Aircraft Wing

**DOI:** 10.3390/biomimetics4030064

**Published:** 2019-09-12

**Authors:** Frédéric Moens

**Affiliations:** ONERA, The French Aerospace Lab, Aerodynamics Aeroelsaticity and Acoustics Department, 92190 Meudon, France; Frederic.moens@onera.fr; Tel.: +33-1-4673-4211

**Keywords:** morphing, droop nose, trailing-edge flap, Natural Laminar Flow wing

## Abstract

This article presents some application of the morphing technology for aerodynamic performance improvement of turboprop regional aircraft. It summarizes the results obtained in the framework of the Clean Sky 2 AIRGREEN2 program for the development and application of dedicated morphing devices for take-off and landing, and their uses in off design conditions. The wing of the reference aircraft configuration considers Natural Laminar Flow (NLF) characteristics. A deformable leading edge morphing device (“droop nose”) and a multi-functional segmented flap system have been considered. For the droop nose, the use of the deformable compliant structure was considered, as it allows a “clean” leading edge when not used, which is mandatory to keep natural laminar flow (NLF) properties at cruise. The use of a segmented flap makes it possible to avoid external flap track fairings, which will lead to performance improvement at cruise. An integrated tracking mechanism is used to set the flap at its take-off optimum setting, and, then, morphing is applied in order to obtain a high-performance level for landing. Lastly, some performance improvements can be obtained in climb conditions by using the last segment of the flap system to modify the load distribution on the wing in order to recover some extended laminar flow on the wing upper surface.

## 1. Introduction: The Different Uses of Morphing Technology for Aerodynamic Performance Enhancement

### 1.1. Use of Morphing Technology for Flight Control

Since the beginning of aviation history, the use of deformable surfaces for controlling flight has been present. The most famous example is the Ader’s Eole airplane in which the design was inspired by an analogy of bat or bird wings (or Leonardo da Vinci drawings) (Figure 1).

Surface shape modification by the use of flexible structures was used for flight control for most of the airplanes at this period (Figure 2). However, due to the increase of flight speed, and, consequently, of the dynamic pressure in flight conditions, these structures appear to be fragile and need to be reinforced, which leads to a dramatic increase of the weight of the deformation system. The use of rigid structures in combination with surface control elements became the standard. Note that, strictly speaking, the use of an aileron for flight control or the deployment of flaps or slats at take-off or landing phases can be considered as “morphing,” which means that the shape of the wing is modified in order to improve its performance for a flight “off design” condition. Currently, a shape is considered morphed if it includes deformation of the initial surface by the use of flexible materials or mechanical systems.

### 1.2. Use of Morphing Technology for Adaptation to Flight Conditions

Later, introduction of morphing technology on military aircrafts has shown significant aircraft performance improvements on a large spectrum of flight conditions. For instance, the use of a variable swept wing in supersonic aircrafts to improve performance at transonic or low speed conditions is a good example (Figure 3).

In the Advanced Fighter Technology Integration (AFTI) program, by NASA and US Air Force, the F-111 wing was equipped with control surfaces so that the airfoil camber was modified and monitored during flight (Figure 4). Flight tests confirmed a significant gain in aerodynamic performance when compared to the reference wing [6].

However, we have to take care when extrapolating potential benefits for transport aircraft applications, for which flight conditions that have to be considered are more limited. For these applications, the introduction of a multi-point multi-disciplinary optimization (MDO) process in the conception has led to highly efficient design and it would be unexpected to see some significant extra gains. For instance, the possibility of a play on load distribution by the variable twist technique in order to match the elliptic span loading distribution is often presented as a good point for the use of morphing technology. Additionally, for transonic transport aircraft for which wing flexibility has to be taken into account, it is known that the optimum span loading considering aero-structural optimization is not elliptic (Figure 5), and is found by MDO processes. Therefore, an elliptical span loading may not be the target to reach for drag minimization.

On the other hand, for subsonic aircrafts with more rigid structures, such as turboprop, the trapezoidal unswept wing shape generates a quasi-elliptical span loading. It is, therefore, very difficult to significantly improve the lift induced drag component for an optimized airplane around its design point.

### 1.3. Use of Morphing Technology for Performance Improvements at off Design Conditions

However, optimization based on fuel consumption and weight minimization generally lead to solutions that are much more sensitive to off-design conditions. The use of morphing technology on wings can help improve performance for these off design conditions (climb, high speed) or to extend the flight domain (buffet alleviation, load control, response to gust), as described in the famous article from Hilbig and Koener [10] or in the book of Concilio et al. [11]. It is also possible to apply morphing technology for surface control such as aileron or on the rudder to replace the current mechanisms based on rotation of a rigid shape.

A final application of morphing technology by using deformable surfaces is noise reduction. It is known that major acoustic sources are located at surface discontinuities (slat and flap ends, see Reference [12]) and the use of a continuous surface will suppress the noise emission at these locations. For instance, tests carried out by NASA on a business jet configuration (Figure 6) will certainly show significant noise reduction when compared with the reference plane. However, a global performance assessment has to be stated because, for some cases, the existence of discontinuities helps improve aerodynamic efficiency. For instance, for high lift configurations, a slotted flap is much more efficient than a plain flap, and, sometimes, some vortices are created in order to improve maximum lift (slat/fuselage junction or nacelle strakes).

Lastly, at the end of the design process, it is necessary to verify if the gain in aerodynamic performance is not balanced when there is an increase of weight. It is necessary to verify if this is due to the system itself or the structure enforcements.

### 1.4. Use of Morphing Technology at Low-Speed

All the previously mentioned benefits provided by the use of morphing technology on an aircraft are for high speed flight conditions. However, the use of morphing technology at low speed conditions, namely take-off or landing, can also be the source of significant performance improvements. As already mentioned, high lift devices can be considered as belonging to the family of morphing systems, and the performance level obtained by a system made of a single slotted Fowler flap and a slotted leading edge slat is almost the maximum achievable level without active flow control. The drawback is that heavy complex mechanisms are necessary to set the elements at their position. When stowed, some external fairings are considered to hide the mechanics in order to minimize both friction and lift induced drag components in cruise conditions. However, the selection of a high lift system depends on the performance required for take-off or landing conditions, with the main one being the maximum lift and the stall angle. The specificity of high lift systems is that, depending on the needs, one system has to be used [14]. If there is a need to increase the stall incidence, a leading-edge device has to be used. If lift has to be increased at a given flight angle, the use of a trailing edge device is necessary, but, in that case, the stall angle generally decreases. Both systems can be combined for both maximum lift and stall angle increases. For both types of devices, morphing technology can be considered.

Among the different well known leading edge devices, the droop nose is a good candidate for the application of morphing technology on NLF wings by the use of compliant deformable structures [15,16]. For trailing edge devices, the use of twistable segmented flaps can be used to increase the deflection at a fixed global position. Additionally, when the flap is stowed, the last segment can be used in high speed conditions to optimize the wing twist or load distribution. Lastly, if the actuation system can be hosted into the wing airfoil shape without external fairing, a significant drag reduction will be achieved for high speed conditions.

Such an application of morphing technology to improve low speed performance has been evaluated in the framework of the AIRGREEN2 EU funded program. This program considers a regional turboprop aircraft configuration for which Natural Laminar Flow (NLF) technology has been considered for the design. This article presents the main outcomes of using morphing technology for advanced high lift systems designed on this NLF wing in order to reach the aerodynamic performance level required. In the second phase, the use of the flap deformation system in climb conditions has been considered for enhancing performance in this flight condition.

## 2. Design of the Regional Aircraft AG2-NLF Wing

### 2.1. Aircraft Configuration

The reference aircraft considered is a 90-pax turboprop configuration (Figure 7) designed by the Leonardo Company in the framework of the CleanSky 1 Green Regional Aircraft (GRA-ITD) program.

The wing airfoils were redesigned by ONERA at cruise conditions for Natural Laminar Flow capabilities, but the wing planform was not modified. This configuration is referred to as AG2-NLF in the following. The design considered a multi-point optimization of the tip and root airfoils for cruise, climb, and low-speed conditions, in order to have a satisfactory performance level on a large part of the flight domain through an extended natural laminar flow on the upper and lower surfaces. Some details about the NLF wing design are given in Reference [15]. In this case, only the main results are recalled.

First of all, it was considered mandatory to keep the original trapezoidal wing planform unchanged. Then, if necessary, a linear wing twist could be considered in the outer part of the wing, but not in the inboard wing portion. Therefore, the redesign of the wing considered two sections, including one at the root and one at the tip. Lastly, due to the low sweep value, two-dimensional computations have been used for the airfoil design. Three dimensional computations were considered at the final stage for the performance assessment.

### 2.2. Numerical Methods Used

In order to have the possibility to evaluate many airfoils in a reduced time frame, fast numerical methods have been used in the airfoil optimization process, namely the Drela’s ISES/MSES code [17]. This code allows the use of computations at fixed C_L_ in fully turbulent or a free transition mode. The airfoil shape parameterization is based on PARSEC representation [18] considering 10 design variables among the 11 PARSEC’s parameters (trailing edge base thickness was fixed at its initial value). The adopted optimizer is the DOT gradient—based method from Vanderplaats [19] that works to minimize an objective function defined as the sum of drag coefficients at different C_L_ values in a high-speed condition, under some constraints on the original wing structure sizing and fuel tank capacity.

The elsA CFD software [20] (ONERA-Airbus-Safran property), that solves the compressible three-dimensional RANS equations by using a cell-centered finite volume spatial discretization on structured multi-block meshes, has been used for evaluating the aircraft performance. For the spatial scheme, the one proposed by Jameson [21], is used for the conservative variables. A fourth order linear dissipation κ_4_ is generally used, with added second order dissipation terms κ_2_ for treatment of flow discontinuities. In the present study, κ_2_ was set to zero due to the low free stream Mach number, and the fourth order dissipation was set to κ_4_ = 1/16 or 1/32. The turbulence model used is the one equation Spalart-Allmaras model with the QCR modification [22] for computations at cruise conditions, or the two-equation k-ω Menter SST [23] for high-lift configurations. Multi-grid computations have been used for convergence acceleration.

For the performance evaluation of the NLF wing at cruise or climb conditions, it is necessary to compute the location of the transition line on the wing. During the RANS computations, the elsA software has the capability to compute laminar flow regions and to determine the transition location by using the AHD compressible criterion for Tollmien-Schlichting instabilities and the C1 criterion for crossflow instabilities within the iterative convergence process [24].

### 2.3. AG2-NLF Wing Design

For each airfoil, the design considered a multi-point optimization for cruise, climb, and low-speed conditions (Table 1) in order to have a satisfactory performance level on a large part of the flight domain. The Reynolds numbers given in this table are based on the wing aerodynamic chord of 2.565 m.

As we consider 2D optimizations, the C_L_ level to be used for the design point has to correspond to local values observed on the wing at the airfoil location. A tool used for preliminary conceptual design, based on analytical relations, has been used in order to make this correlation. Figure 8 presents the evolution of the local lift coefficient on the trapezoidal reference wing for Cruise and Long-Range conditions given in Table 1. It can be seen that, for the root airfoil, the local C_L_ to be considered is in the same order of magnitude as the aircraft lift coefficient (green curves), whereas, for the tip airfoil, the 2D values correspond to about 50% of the 3D ones.

The numerical optimization process considered a composite cost function made of the sum of drag coefficients at different C_L_ values for which a large natural laminar flow extent is expected (Table 2).

Figure 9 presents the two-dimensional computed performance of the re-designed root and tip airfoils of the AG2-NLF wing at nominal cruise conditions (M = 0.52, Altitude = 20,000 ft). Performances of the reference (turbulent) airfoil at the same conditions are indicated. The new airfoils exhibit NLF characteristics on a large range of local C_L_ around the design value, which leads to a significant friction drag reduction. In addition, the performance of these airfoils in turbulent conditions are similar (a tip) or event better (at root) than for the reference turbulent one.

Then, the 3D wing was generated considering these two airfoils. The twist was adapted in order to take low speed performance into account (Figure 10). The question of the stall onset was investigated by introducing a twist in the outboard wing in order to displace the wing critical section that is more inboard. The stall onset is driven by the pressure gradient at the wing leading edge, which is characterized in the first order of magnitude by the absolute level of the minimum pressure coefficient. Figure 10b compares the computed spanwise Cp distributions for the reference untwisted wing and for the optimized twisted wing at low speed conditions and α = 15°. Only the portion of the wing between the span wise position Y = 5 m (right) and the tip (left) are presented. The location of the critical wing section, where the stall onset should appear, is marked with a black circle and the location. The aileron area is between the dashed line and the tip. It can be seen that, for the untwisted wing, the critical section is located within the aileron area, which is not acceptable. The introduction of a twist of 4° at the tip section moves the critical section inboard at an acceptable spanwise position and has been retained.

It was verified that this linear twist has no major impact on the computed laminar flow extent on both surfaces at the design point at cruise (Figure 11).

In the second phase of the project, some high-lift devices have been designed and adapted to this wing. Considering the high level of performance required at low speed, the use of morphing technology was mandatory for both leading edge and trailing edge devices.

## 3. Leading Edge Device: Use of a Morphed Droop Nose

Figure 12 presents the computed pressure distribution at low speed conditions (M = 0.15, Altitude = 0 ft) for the clean AG2-NLF wing case. It can be seen that a significant pressure peak is found on the wing at high incidence. This is a common behavior observed for the wing designed in order to have laminar flow characteristics at cruise conditions. In that case, the airfoil leading edge radius is reduced when compared to a turbulent one, in order to drive the favorable pressure gradient to maintain the flow laminar. The drawback is that, at high angles of incidences, a strong acceleration is found at the airfoil leading edge, which will increase the risk of leading edge stall occurrence. It is, therefore, necessary to use a leading edge device in order to act on the pressure peak at low speed conditions. Lastly, this device has to be compatible with the constraint of keeping laminar flow at cruise conditions on the wing when not deployed. The morphed droop nose device was retained.

Compared with a standard droop nose, a morphed droop nose allows the redesign of the baseline wing shape, which can be optimized by considering only the flight conditions that do not require shape changes introduced by the morphing. This aspect provides an additional advantage in terms of the aerodynamic benefit because different external shapes can be defined to optimize the aerodynamic performances in different flight conditions. The different shapes can be designed separately considering that the morphing allows the transition between them to preserve the shape continuity and avoids any type of step and gap. This advantage is greater in the case of the laminar wing where the NLF wing can be optimized for the high-speed conditions and the same wing, equipped with the morphing droop nose, for the low-speed conditions.

The detailed process considered for the design of the droop nose adapted to the AG2-NLF wing (Figure 13) is detailed in References [15,25]. It considered aero-structural optimizations carried out by Politecnico di Milano and aerodynamic performance assessments were done by ONERA by using CFD.

First, a preliminary performance assessment at low speed has been done in a two-dimensional flow for a pre-designed landing configuration considering a standard flap. Figure 14 presents the computed C_L_(α) curves for a geometry considering a droop nose or not. As for any leading edge devices, the use of a droop nose leads to an increase of maximum lift and stall angle, but with nearly no effect on the lift level for lower incidences. Values indicated for the gains (+11.5% in C_Lmax_ and + 4 degrees in stall angle) are for information only, since they are based on a 2D airfoil, and not on the 3D wing.

Different optimized droop nose shapes from PoliMi works were compared, which led to the selection of one geometry that has been adapted to the 3D wing-body configuration for a CFD evaluation of the performances by ONERA. Figure 15 compares the pressure distributions computed on the AG2-NLF airplane at take-off conditions (M = 0.20, Altitude = 0 ft) for an incidence of 12.5°. The use of a droop nose significantly decreases the suction peak at the leading edge, which makes the pressure gradient less favorable for a leading edge stall occurrence. Therefore, the stall will occur at a higher incidence, as observed in Figure 16a.

There is another (favorable) effect observed on drag (Figure 16b). The change in pressure distribution at the wing leading edge due to the droop nose deflection leads to a constant decrease in the drag coefficient, which corresponds to a reduction of about 5.5% at the flight condition for the wing-body configuration.

## 4. Trailing Edge Device: Use of a Multi-Segmented Flap System

Among the different possibilities to deform a wing for performance improvement, one considers the multi-functional wing trailing edge. Such a system acts on local wing shape deformations in order to modify the span load distribution. The morphing system considered is similar to the one designed for an aileron in the framework of the CRIAQ project [26] but is adapted to a trailing edge flap. Detailed information about the design of the reference flap system used on the AG2-NLF wing are given in References [15,27]. However, its spanwise extension corresponds to the place dedicated to the trailing edge flaps (Figure 17). This system has to, therefore, be integrated to the flap, which leads to geometrical constraints for the design of the high-lift system. Considering the multi-segmented flap system retained, the use of the last segment as a morphing device when the flap is stowed leads to a maximum shroud location at 92.5% on the wing’s upper surface. Location of the cove on the lower surface is driven by the wing structure’s rear spar location.

As mentioned in Reference [15], it is possible to find an optimal aerodynamic setting for landing conditions when considering the rigid flap shape used for take-off. In the framework of the AIRGREEN2 program, different flap deployment progression laws were investigated by Siemens [28,29], who was responsible for the flap actuation system in the project, and it was evaluated numerically by ONERA. The outcome is that it was not possible to find a mechanism that will ensure take-off and landing settings that will not need an external fairing for this rigid flap shape. However, it was possible to design a fully integrated tracking system to set the flap at the optimized take-off configuration. Next, the possibility to deform the flap shape was investigated. It was observed whether the deformation took place due to morphing with the flap located at its take-off settings in order to obtain “sufficient” aerodynamic performance for landing conditions (Figure 18). Note that it is not evident that such a process would necessarily work, as we start from a take-off setting and shape (for the front flap segment). These are parameters that usually need to be modified for landing conditions (differences between red and blue shapes in Figure 19.

Figure 20 presents the morphing flap system from UniNa adapted to the AG2-NLF wing geometry. Note that, in the UniNa design, the different hinge lines are parallel to the flap trailing edge, and not at a constant local chord. This means that, when deformation is applied, the flap shapes are different at each spanwise location and that the performance evaluation considering a 2D wing section is not possible. Three-dimensional numerical CFD evaluations are mandatory.

Due to mechanical constraints, there are some physical links existing between hinge 1 and 2, which lead to the kinematic law presented in Figure 21 for the rotation angles between these two hinges. For instance, it means that there is a 5° deflection at hinge 1 applied that leads to 15° at hinge 2 as a global deflection value (or +10° applied at hinge 2 after the 5° deflection for hinge 1). Deflection values for hinge 3 are free, but limited to 10° in amplitude. The symbols correspond to the configurations that have been evaluated numerically by ONERA.

Preliminary studies carried out based on the rigid flap shape gave an optimum flap deflection around 35° for landing. Taking into account the initial flap deflection of 20°, corresponding to the take-off case, we have to investigate configurations with a deflection angle of the second hinge around 15°. Based on results presented in Figure 22, the best combination for maximum lift optimization corresponds to a deflection of 15° for the second hinge, and a 10° extra deflection for the last segment.

The final performance assessment for the landing configuration (M = 0.15, Altitude = 0 ft) considered the use of the droop nose designed previously in combination with the deformable flap. Figure 23 compares the computed performance for both cases. It can be seen that the combined use of these two morphing devices leads to a significant improvement in both C_Lmax_ and the stall angle. The requirement in terms of the C_Lmax_ level for landing conditions is respected, whereas it is not reached for the configuration equipped with the standard leading edge. A final verification considered the stall process of the wing equipped with the droop nose. It was verified whether there is no separation onset in the aileron area for flight control considerations. Figure 24 presents the computed skin friction lines for the landing configuration with the standard leading edge. It can be seen that a separation occurs on the completed wing’s upper surface at stall. Figure 25 presents similar plots for the configuration equipped with the dropped nose. Stall occurs more gradually, and starts from the wing-body junction.

## 5. Use of the Flap Morphing System for Performance Improvement in Climb Conditions

When considering an NLF wing at off-design conditions with high C_L_ values, the change of the pressure gradient on the wing upper surface becomes less favorable to maintain laminar flow on a large chord extension. It can be proposed to morph the airfoil shape in order to recover NLF characteristics, as investigated in the task of the CRIAQ project [30,31]. For the AG2-NLF regional airplane, the multi-functional twistable trailing edge could help recover the laminar extent by an adaptation of the pressure gradient in the off-design condition [27]. Considering the C_L_ related to a high-speed climb condition (M = 0.36, Altitude 15,000 ft, C_L_ = 0.85), free transition computations show that laminar flow on the upper surface is somewhat lost on the outer wing. The possibility to deflect the last segment of the multi-functional flap in order to improve performance in these conditions was investigated. Different tab deflections have been considered (2.5°, 5°, 8°, and 10°). For the performance evaluations by CFD, the surface grid used for cruise evaluation has been deformed in the tab region and a mesh deformation technique, which is similar to the one used in the SARISTU project [32] and described in Reference [11]. Figure 26 shows such a configuration with a tab deflection of 10°.

Figure 27 presents the computed lift over drag ratio (LoD) for the wing-body configuration of the AG2-NLF airplane for climb conditions. The black curve corresponds to the performance of the reference wing. The color lines correspond to the different configurations with the tab deflected (“delta” corresponds to the tab deflection). For these conditions, a loss of performance is observed due to the loss of laminar flow on the wing upper surface (Figure 28a). The use of small tab deflections (2.5° or 5°) allows for the recovering part of the laminar flow on the wing’s upper surface (Figure 28b) and shifts the LoD curve to higher C_L_ values that increase the performance by about 2%. However, higher deflection angles (8° and 10°) lead to a global decrease of performance. When considering the drag breakdown between friction and pressure components (Figure 29), it can be seen that, if an increase of tab deflection leads to a continuous decrease of friction drag, there is an increase of pressure drag that leads to an optimum value for low tab deflections.

Lastly, Figure 30 compares the different wing span load evolutions at the design point in climb conditions for the different configurations considered. Applying a deflection to the multi-functional flap system has an effect on the span load evolution but only in the portion where the system is located. Recovering an elliptical span loading would mean to act on the outer part of the outer wing, where the aileron is present. Therefore, gains on the lift induced drag are not possible if there is no action in this area, through an aileron deflection (or morphed aileron) or a spanwise extension of the multi-functional flap system up to the tip.

## 6. Conclusions

Aerodynamic performance for take-off and landing phases of a regional turboprop configuration equipped with an NLF wing have been significantly enhanced by the application of morphing technology for high lift devices. The use of a deformable morphing-based droop nose, designed by PoliMi, as a leading edge device has been considered since it preserves the surface quality when retracted in cruise conditions. This device leads to an increase of both C_Lmax_ and the stall angle.

For the trailing edge device, a multi-segmented flap has been considered. For low speed applications, the objective was to obtain a mechanism that will not require any external fairing, which will significantly improve the drag at cruise. Different strategies have been considered in an interactive process between the partners involved (namely UniNa for the segmented flap system, Siemens for the definition of the tracking system, and ONERA for the aerodynamic performance assessment). It was found that an integrated tracking system was possible to set the flap at its take-off optimum location. Then, morphing was applied on the flap in order to reach the performance required for landing conditions.

The possibility to use the flap’s last segment as a morphing device for performance improvements in climb conditions has been verified. However, this performance improvement was obtained by reducing the friction drag, thanks to an adaptation of the laminar flow on the wing upper surface to flight conditions. It is, therefore, not definite that such a performance improvement can be found when considering turbulent wings.

In this article, we focused on aerodynamic performance improvements for the wing-body reference configuration. Each component (droop nose, flap system) has to be optimized in order to take into account the weight balance, the system complexity, and aeroelastic behavior, and to be integrated into the complete aircraft architecture. Then further design phases can start by considering the propulsion system (nacelle, engines, and propellers), the control surfaces (ailerons, horizontal tail, fin), the mechanical components (track systems), the structure, and the energy sources. All these elements have to be integrated and considered for a complete aircraft performance evaluation on the complete flight envelope.

Lastly, the use of morphing technology is not restricted to pure aerodynamic performance improvements. The use of deformable structures for load control during flight is another important application of this technology (see Reference [33]) and this was their first use in aviation history.

## Figures and Tables

**Figure 1 biomimetics-04-00064-f001:**
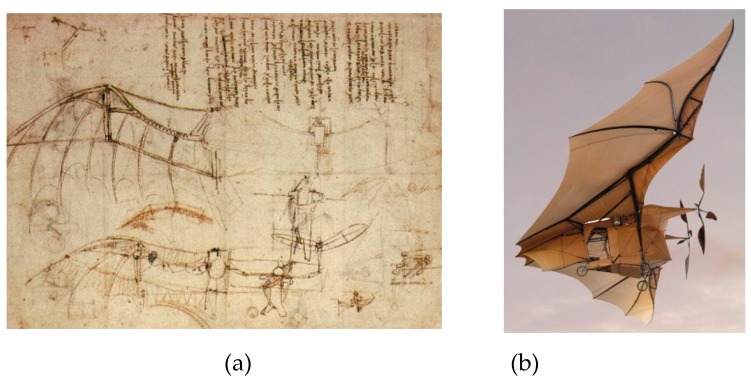
How to fly? First ideas. (**a**) Leonardo da Vinci drawings [1]; (**b**) Clément Ader’s *Avion III* [2].

**Figure 2 biomimetics-04-00064-f002:**
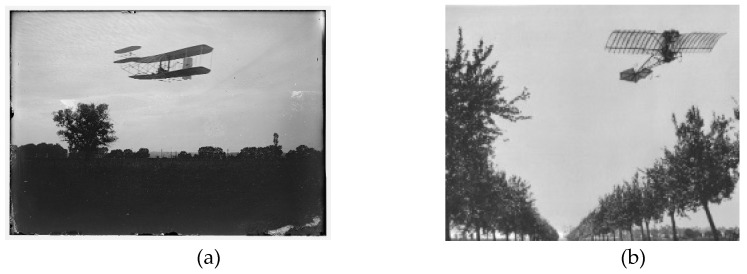
Pioneer ages—Use of morphing for the flight control surface. (**a**) Wright brothers’ *Flyer* [3], (**b**) Santos-Dumont’s *Demoiselle* [4].

**Figure 3 biomimetics-04-00064-f003:**
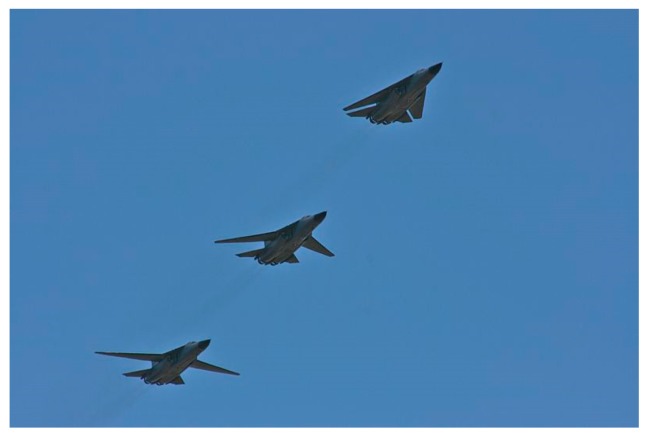
F-111 Aircraft wing sweep modification sequence [5].

**Figure 4 biomimetics-04-00064-f004:**
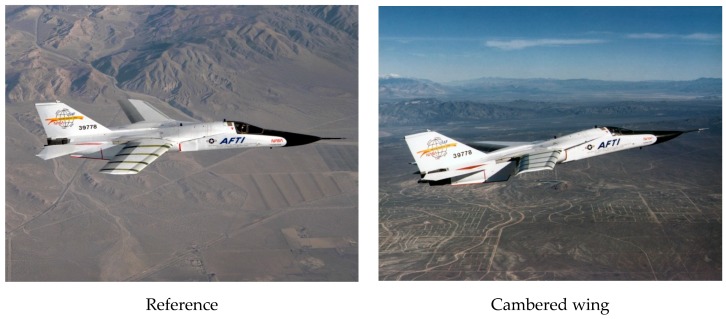
AFTI/F-111 aircraft in flight [7,8] with a variable camber wing.

**Figure 5 biomimetics-04-00064-f005:**
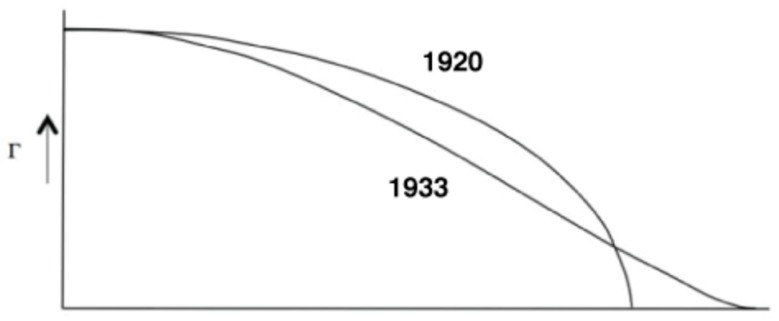
Optimal span load distribution for minimum drag [9] from Prandtl’s studies. No constraints: elliptic shape (1920)—Wing with the same structural weight: bell-shaped (1933).

**Figure 6 biomimetics-04-00064-f006:**
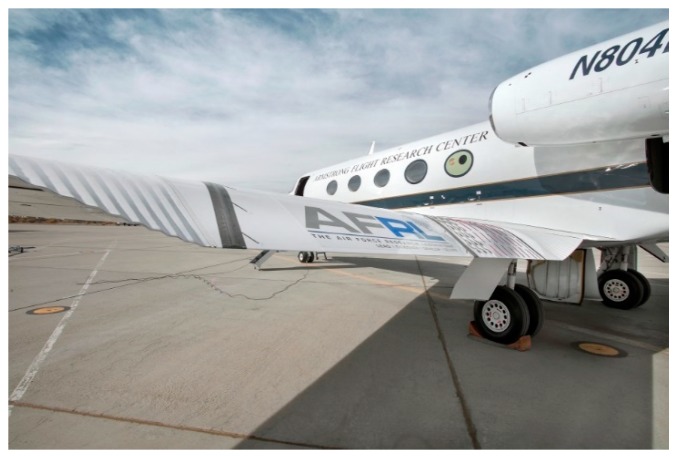
Adaptive Compliant Trailing Edge (ACTE) flaps on NASA’s Gulfstream III aeronautical test bed [13].

**Figure 7 biomimetics-04-00064-f007:**
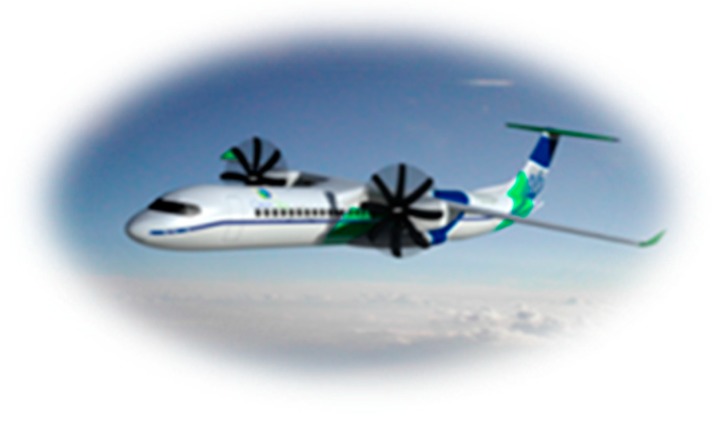
Reference TP90 aircraft (picture from Leonardo—with permission).

**Figure 8 biomimetics-04-00064-f008:**
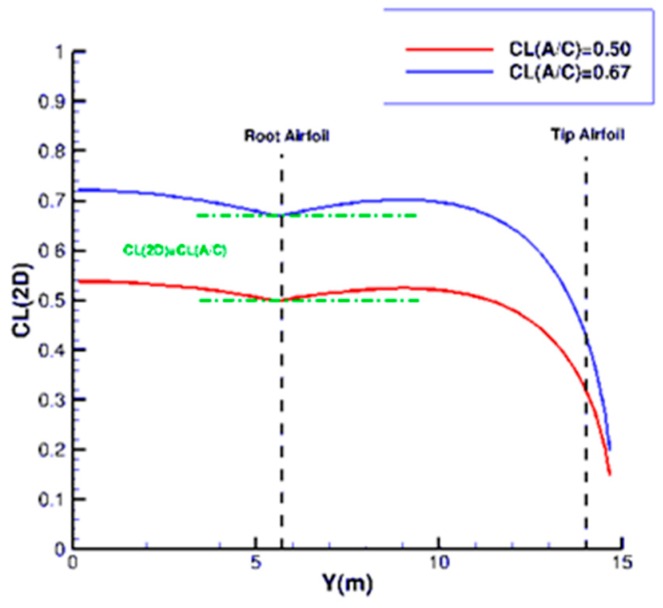
Span wise evolution of the local lift coefficient corresponding to Cruise and Long-Range conditions.

**Figure 9 biomimetics-04-00064-f009:**
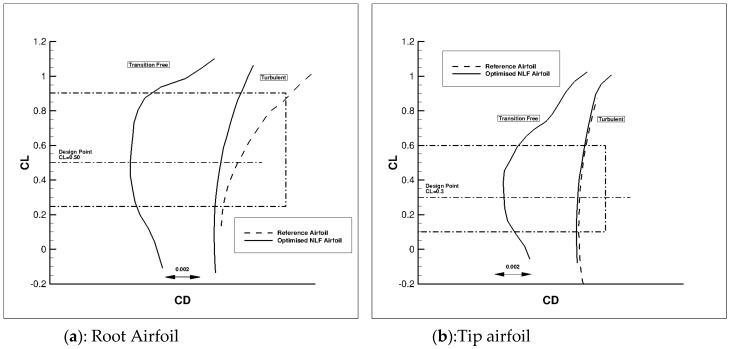
Root (**a**) and tip (**b**) airfoil performance of the AG2-NLF wing at cruise conditions (M = 0.52, Altitude = 20,000 feet).

**Figure 10 biomimetics-04-00064-f010:**
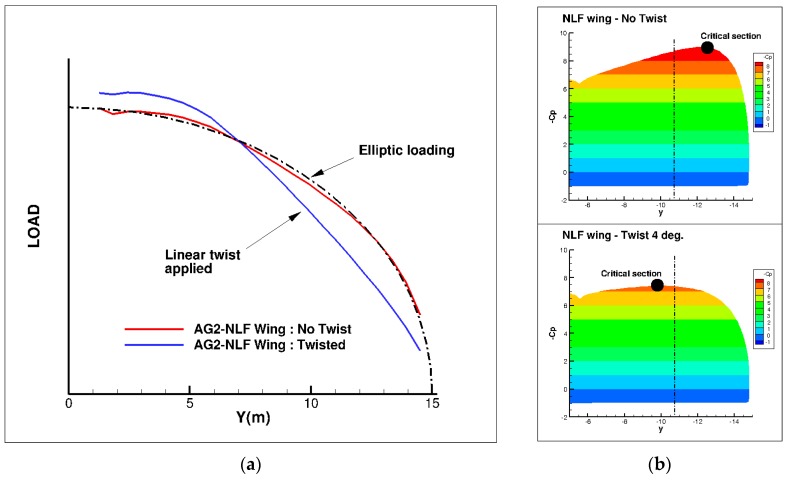
Effect of a linear twist on the wing. (**a**) Computed wing span load evolution of the AG2-NLF wing at nominal cruise flight conditions (M = 0.52, Altitude = 20,000 ft, C_L_ = 0.50). (**b**) Minimum pressure level at low speed conditions (M = 0.20, Altitude = 0 ft, α = 15°).

**Figure 11 biomimetics-04-00064-f011:**
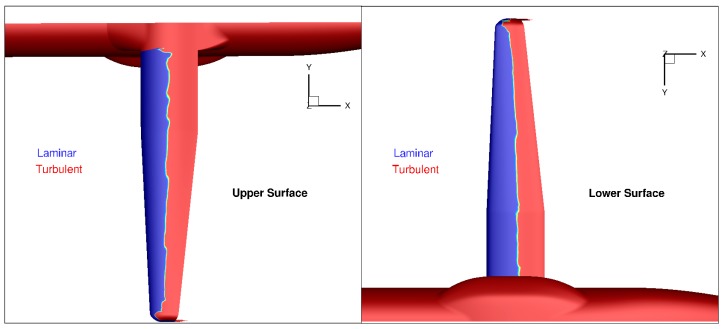
AG2-NLF wing at nominal cruise conditions (M = 0.52, Altitude = 20,000 ft, C_L_ = 0.50)—Computed extension of laminar flow on the wing surfaces (elsA computations).

**Figure 12 biomimetics-04-00064-f012:**
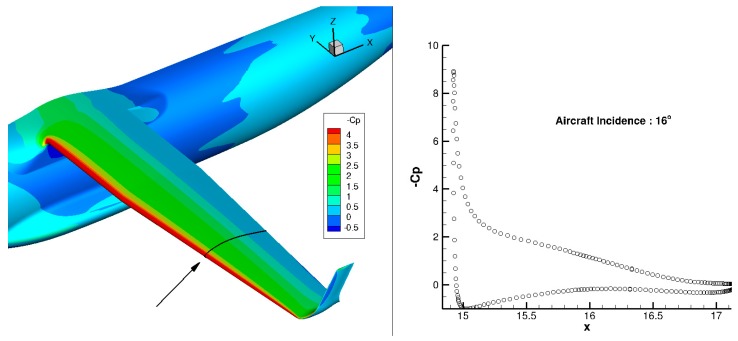
AG2-NLF wing at low speed (M = 0.20, Altitude = 0 ft): development of a high suction peak at the leading edge.

**Figure 13 biomimetics-04-00064-f013:**
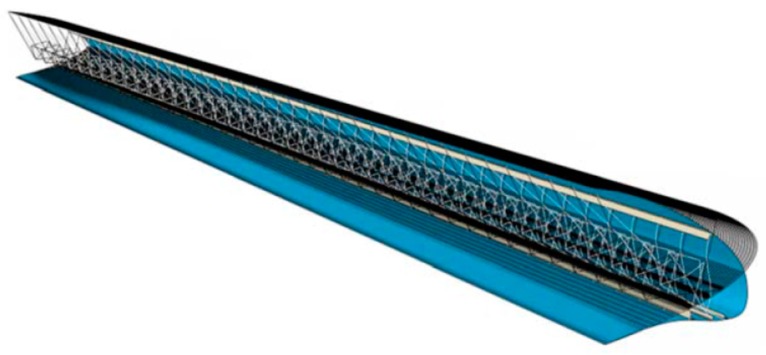
Final 3D droop nose designed by PoliMi [25] with permission.

**Figure 14 biomimetics-04-00064-f014:**
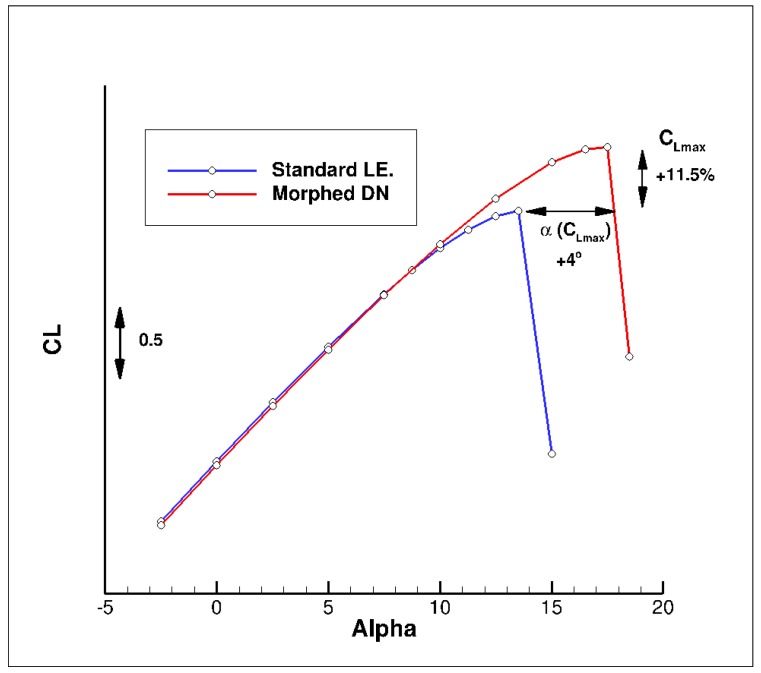
Use of a morphed droop nose device—2D Evaluation of C_L_(α) curves—Landing conditions (M = 0.15, Altitude = 0 ft).

**Figure 15 biomimetics-04-00064-f015:**
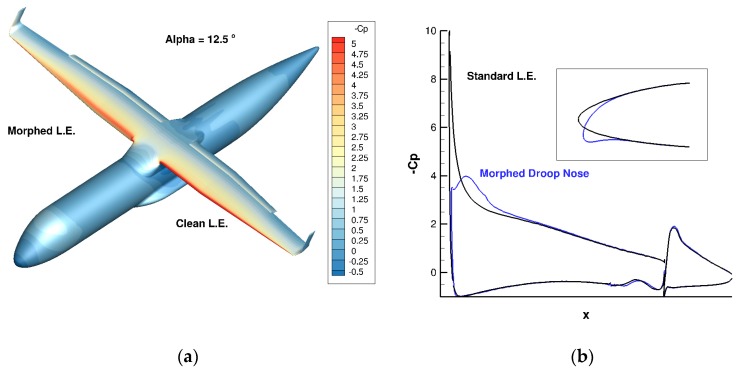
Effect of a droop nose on pressure distribution (Take-off conditions: M = 0.20, Altitude = 0 ft, α = 12.5°). Pressure distribution on the wing (**a**) and at the outboard flap section (**b**).

**Figure 16 biomimetics-04-00064-f016:**
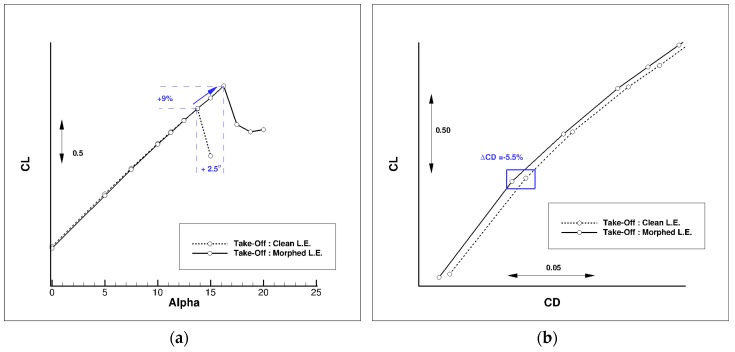
Effect of a droop nose on performance at take-off conditions (M = 0.20, Altitude = 0 ft). (**a**) C_L_(α) curve, and (**b**) C_L_(C_D_) curve.

**Figure 17 biomimetics-04-00064-f017:**
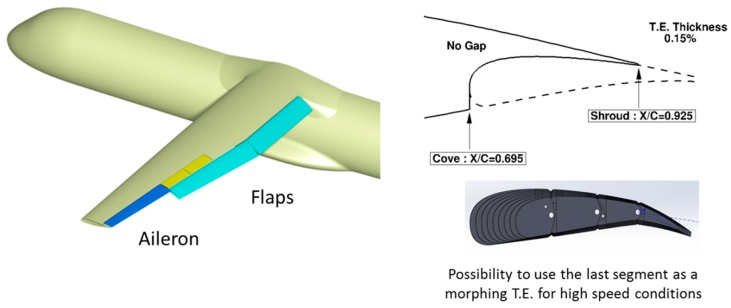
General layout of the AG2-NLF wing for flap arrangements and flap design constraints.

**Figure 18 biomimetics-04-00064-f018:**
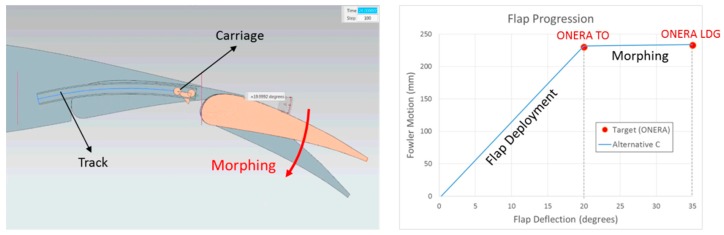
Best alternative for a solution with no external fairing: Take-off configuration, and then apply morphing for landing (Siemens—with permission).

**Figure 19 biomimetics-04-00064-f019:**
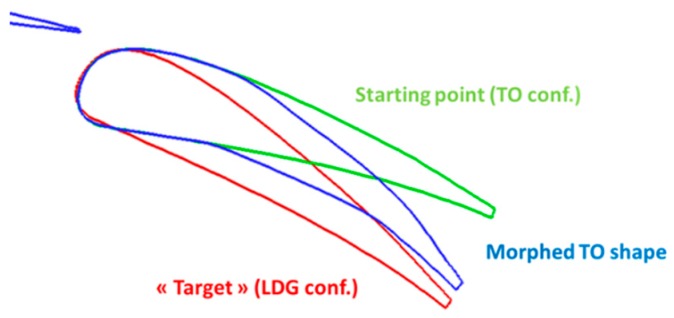
Application of morphing technology on the flap system.

**Figure 20 biomimetics-04-00064-f020:**
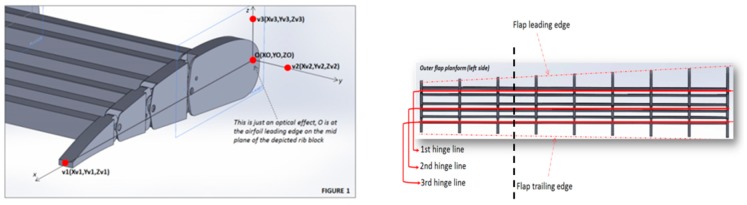
Trailing edge morphing flap. General layout from UniNa (with permission).

**Figure 21 biomimetics-04-00064-f021:**
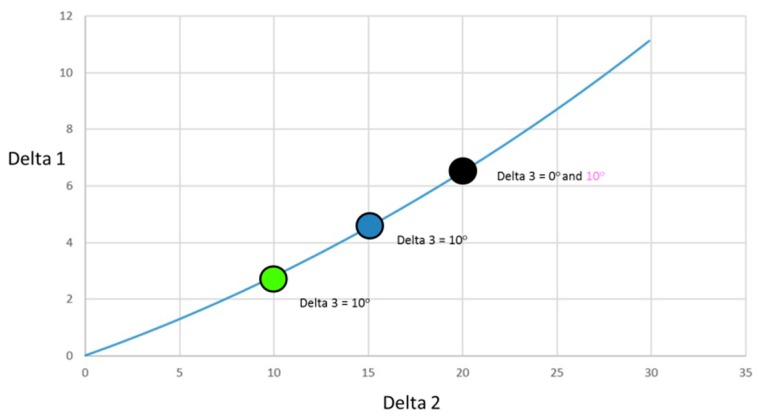
Morphing mechanism: Kinematic law for the rotation values at Hinge 1 and Hinge 2. Rotation at Hinge 3 is free. Symbols correspond to configurations considered for morphed flap at Landing conditions. Global geometrical angles.

**Figure 22 biomimetics-04-00064-f022:**
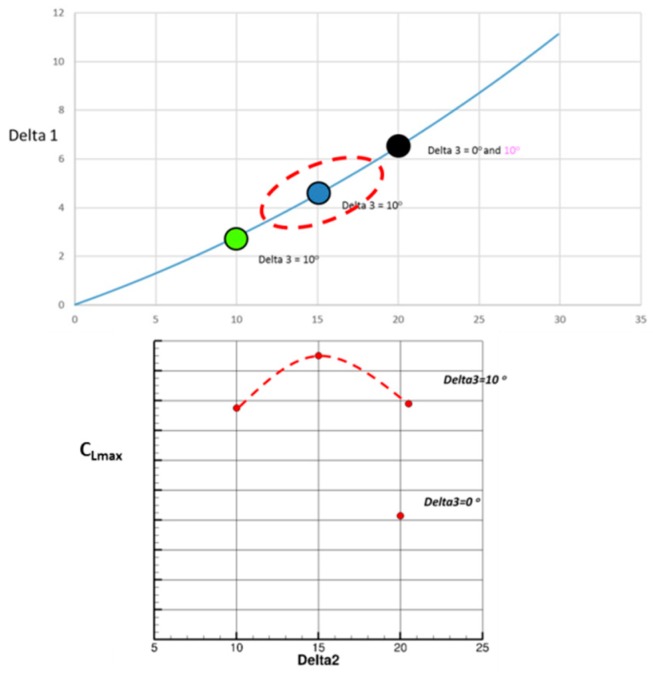
Optimization of the flap morphing system for landing.

**Figure 23 biomimetics-04-00064-f023:**
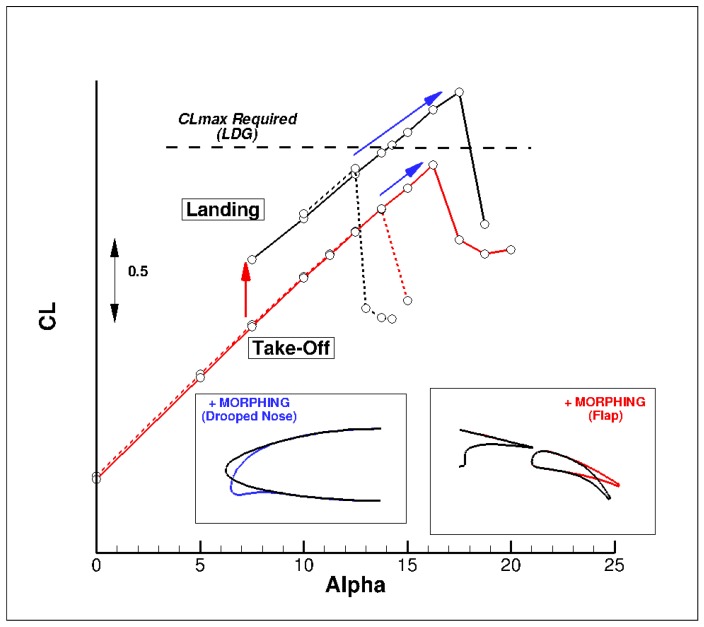
High lift performance of the AG2-NLF equipped deformable elements (droop nose and multi-segmented flap system). Landing: M = 0.15 Altitude = 0 ft—Take-Off: M = 0.20, Altitude = 0 ft.

**Figure 24 biomimetics-04-00064-f024:**
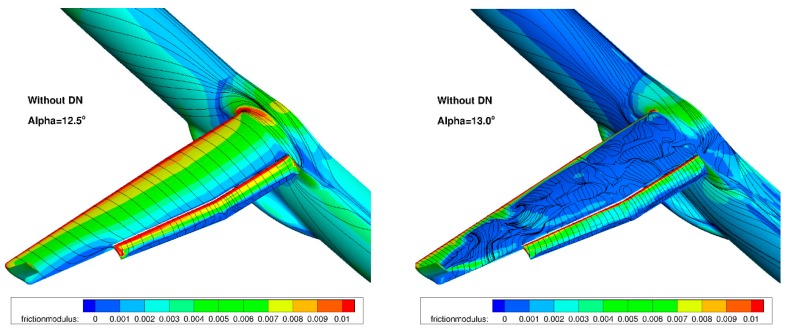
Landing configuration: stall process with standard leading edge.

**Figure 25 biomimetics-04-00064-f025:**
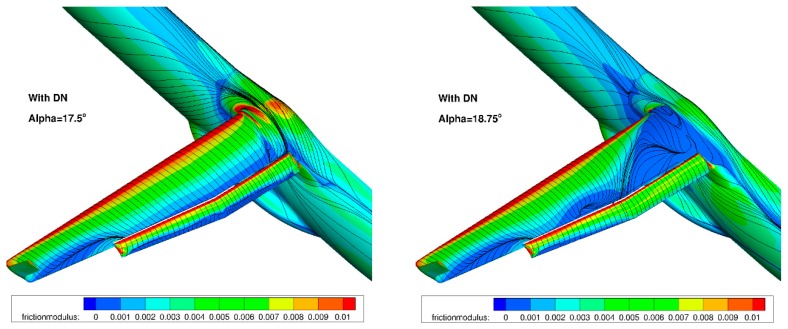
Landing configuration: stall process with the droop nose leading edge device.

**Figure 26 biomimetics-04-00064-f026:**
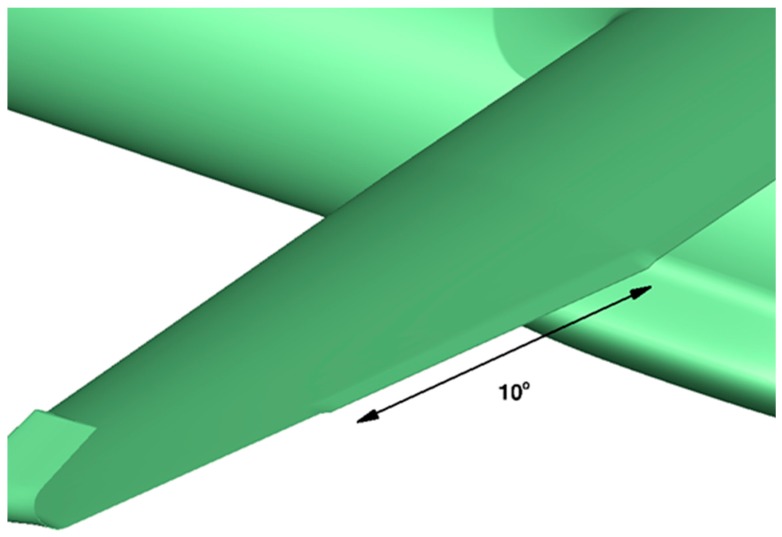
Configuration considered for a multi-functional flap at climb conditions (example shown: tab deflection of 10°).

**Figure 27 biomimetics-04-00064-f027:**
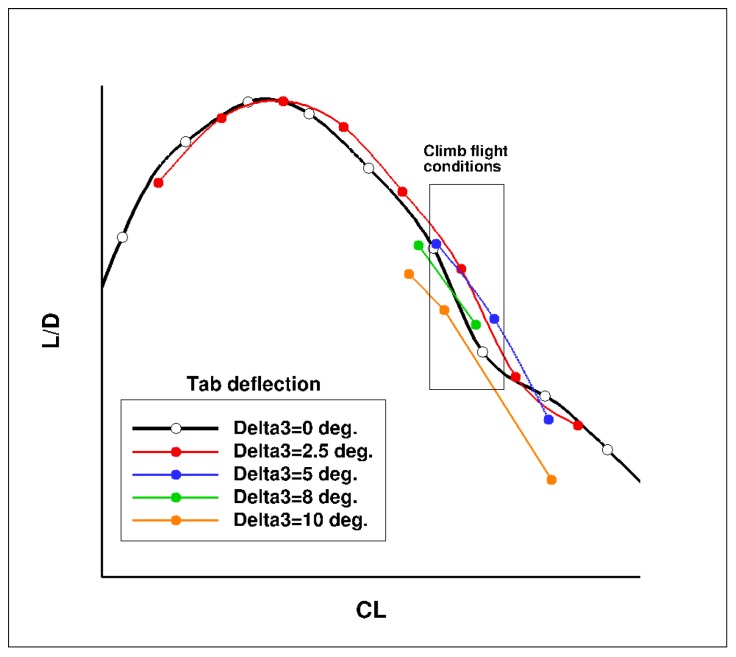
Performance of the multi-functional trailing edge flap (climb conditions—M = 0.36, Altitude = 15,000 ft).

**Figure 28 biomimetics-04-00064-f028:**
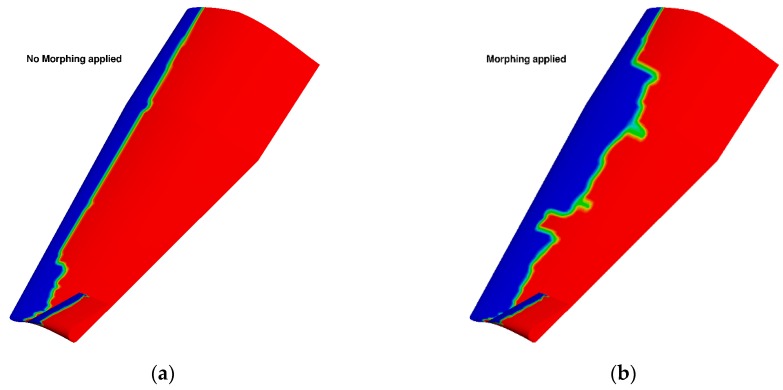
Natural laminar flow extent on the wing’s upper surface for climb conditions (M = 0.36, Altitude = 15,000 ft). (**a**) Reference wing (no morphing). (**b**) Morphing applied (2.5° deflection). Laminar flow in blue and turbulent flow in red.

**Figure 29 biomimetics-04-00064-f029:**
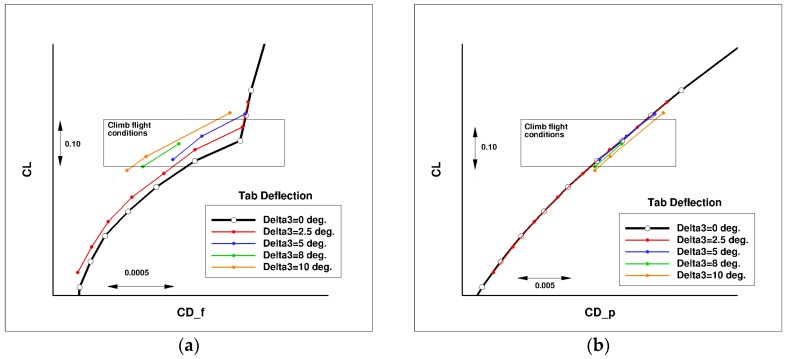
Drag breakdown for the different configuration of multi-functional trailing edge flap (climb conditions—M = 0.36, Altitude = 15,000 ft). (**a**) Friction drag and (**b**) pressure drag.

**Figure 30 biomimetics-04-00064-f030:**
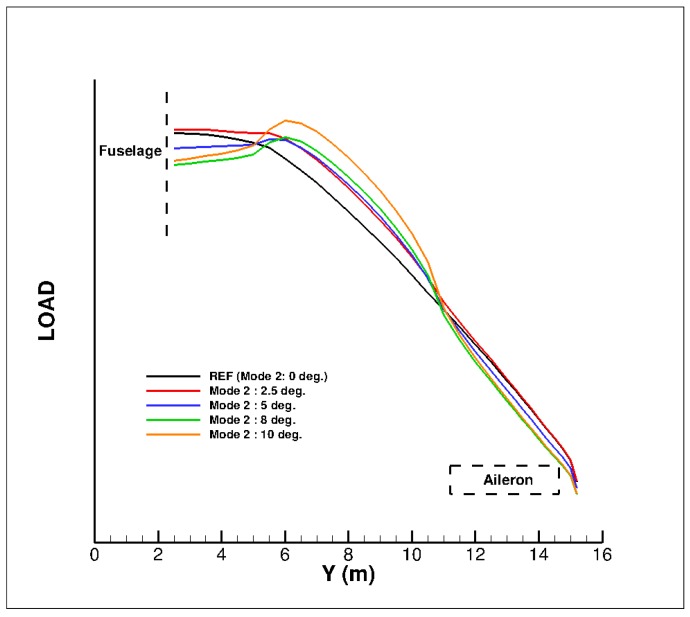
Span loading evolution with a multi-functional trailing edge flap deflected in climb conditions (M = 0.36, Altitude = 15,000 ft, C_L_ = 0.85).

**Table 1 biomimetics-04-00064-t001:** Flight conditions considered for the NLF wing design.

	Mach Number	Altitude	Reynolds Number	CL (Wing + Body)
Cruise	0.52	20,000 ft	17.3 × 10^6^	0.50
Long-Range	0.45	20,000 ft	15.0 × 10^6^	0.67
Low Speed	0.20	0 ft	11.8 × 10^6^	-

**Table 2 biomimetics-04-00064-t002:** Range in local C_L_ coefficients considered for the airfoil multi-point optimization.

	Lower C_L_	Nominal C_L_	Upper C_L_
Root Airfoil	0.25	0.50	0.90
Tip Airfoil	0.10	0.30	0.60

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
