# Peer review of "Augmented Aircraft Performance with the Use of Morphing Technology for a Turboprop Regional Aircraft Wing"

_biomimetics, 2019, doi:10.3390/biomimetics4030064_

Round 1
Reviewer 1 Report
Augmented Aircraft Performance by the use of Morphing Technology for a Turboprop Regional Aircraft Wing
Biomimetics
In this paper, the authors present some application of the morphing technology for aerodynamic performance improvement of turboprop regional aircraft. Some results have been shown based on the framework of Clean Sky 2 REG-IADP AIRGREEN2 program. The paper is well-written on a topical area but some further comments are given here: This paper firstly introduces some literature review and then presents some technical results, both of these two parts makes a great deal in this paper. So it is unclear to read it as a review paper or a technical paper, if it is a review paper, I think the cited reference is too few to support the morphing details. Or if it is a technical paper, reader is hard to realise the highlight of this paper. I suggest some more details can be added or modified in this paper. Furthermore, some high quality papers in morphing aircraft maybe worth to refer.
Morphing Aircraft:
1. A Review of Morphing Aircraft.
2. Morphing Aircraft Systems: Historical Perspectives and Future Challenges.
Leading edge device, Trailing edge device and new technology:
3. Comparison of Constrained Parameterisation Strategies for Aerodynamic Optimisation of Morphing Leading Edge Airfoil
4. Spiral Pulley Negative Stiffness Mechanism for Morphing Aircraft Actuation
5. Conceptual-level evaluation of a variable stiffness skin for a morphing wing leading edge
6. Wind tunnel testing of the fish bone active camber morphing concept
7. Three-dimensional design of a large-displacement morphing wing droop nose device.
8. Extremely deformable morphing leading edge: optimization, design and structural testing.
9. Compliant structures-based wing and wingtip morphing devices.
…etc.
It could be very useful to frame these high quality contribution in the literature.
Author Response
Reply to reviewers’ comments.
First of all, I would like to thank the different reviewers for reading the proposed article and for their comments.
It was suggested to use the “modification” mode of MS-Word in order to highlight the different modifications done. However, as there were some changes in the text/figures/references, it became quite difficult to have a correct vision of the revised version. Therefore, I made the different changes visible using a new color for the text (in red).
Rev 3 suggested adding a list of symbols: done
Rev 2 asked to check permission to use pictures. This was done and the references updated.
• Note : some pictures have been changed compared to first version:
o Use of public domain pictures for Fig 2.
o Original NASA pictures for Fig 4 and 6.
o I have removed the “old” Fig 6 (text is sufficient I guess) and “old” Fig 8. In that case, it is not because of permission, but to avoid misunderstanding: some readers may think that these aircrafts have “problems” which is not the case. It was only for illustration of the size of flap track fairings.
o The permission to use of pictures from AG2 partners has been obtained. All the other pictures have been generated by Onera in the framework of AG2 project.
Rev 2 and Rev3 asked some details of the wing design. As the paper is related to morphing technology, I made a summary of the wing design in the first version. New chapters describing the wing design and the numerical methods used have been added.
Rev2 asked to add the flight conditions for each figure caption: done. “Sea level” has been replaced by “Altitude=0ft”.
Rev 2: explanation of “Delta” on last figures. It was the deflection angle of the last tab segment. For homogeneity with flap system, it has been replaced by “Delta3” and “Tab Deflection” has been added in the figures.
Some words about CRIAQ works have been added, as suggested by Rev 3.
Rev 3: what about testing in wind tunnel ?
There is a follow on project considering wind tunnel tests (GRETEL) about technologies investigated in AG2 (not only High-lift systems, but also new devices for load control). The “problem” is that the planform selected is not the AG2-NLF aircraft, but the reference aircraft design in Clean Sky 1. There were different arguments for that, the main one being that a complete aircraft model is available. The project leader considered that it was possible to adapt the different techno evaluated to the original wing. In my opinion, this is right except for the flap system that has to be redesigned. And taking into account the fact that ONERA is not involved in GRETEL project (except for some tests), I
didn’t make some comments in my paper. However, maybe contributors to other chapters of the Special Issue, dealing with load control, could speak about it.
Rev 1: Apparently, it was not clear if the paper was a review paper of a technical one. The paper is a technical one, but to introduce the work done in AG2, I made an introduction that may be a little bit too long … But I think it is necessary in order to introduce how morphing technique lead to aerodynamic improvements on this configuration, which is not the usual way we have in mind when considering morphing. In general, morphing technologies are investigated at high speed conditions. In this study, the use of morphed flap is mandatory if we want to respect the requirement to have no external fairings for flaps, and the system defined can be used in off design conditions to extend the flight domain.
As Rev2 and Rev3 considered the introductions as OK in their comments, I have not changed its structure, but I have added sub-headers for each applications. I hope it helps for the readers understanding.
Finally, the paper focus on aerodynamic performance improvement thanks to morphing technology. Not the details about how the morphed shape is obtained, except if it leads to a constraint for the design (for instance: how the multi functional flap system is segmented). I guess that all these details will be provided by other contributors to the special issue as there are dedicated chapters for system design, materials, structures etc … Therefore, all the suggested reference will be considered in these chapters.
MOENS Frédéric
ONERA
Reviewer 2 Report
This paper presents a very interesting research in the field of morphing real aircraft technology in the Clean Sky program. Please find here some suggestions that could be used to improve this paper:
1) Please verify all figures - as there is the need of ''permission'' from the authors for their consideration in this paper - even if figures were published on internet.
2) Page 5 - The NLF wing design was given in ref. 16 - Believe that there is the need to add the details of this design in the paper (software, algorithm, etc.).
3) Believe that the ''multi-point optimization of the tip and root airfoils'' would be applied to a wing with airfoils linearly distributed between tip and root airfoils. Please explain more the wing design.
4) Please add on each figure caption (where needed, and start with Figure 14) the flight conditions (Mach number, angle of attack, altitude). Have noticed that in most of the text and on figures, ''sea level'' was mentioned as ''altitude".
5) Please explain better and more the ''Delta'' on Figure 21, 22,27, 29.
Thanks in advance for these minor revisions.
Author Response
Reply to reviewers’ comments.
First of all, I would like to thank the different reviewers for reading the proposed article and for their comments.
It was suggested to use the “modification” mode of MS-Word in order to highlight the different modifications done. However, as there were some changes in the text/figures/references, it became quite difficult to have a correct vision of the revised version. Therefore, I made the different changes visible using a new color for the text (in red).
Rev 3 suggested adding a list of symbols: done
Rev 2 asked to check permission to use pictures. This was done and the references updated.
Note : some pictures have been changed compared to first version:
o Use of public domain pictures for Fig 2.
o Original NASA pictures for Fig 4 and 6.
o I have removed the “old” Fig 6 (text is sufficient I guess) and “old” Fig 8. In that case, it is not because of permission, but to avoid misunderstanding: some readers may think that these aircrafts have “problems” which is not the case. It was only for illustration of the size of flap track fairings.
o The permission to use of pictures from AG2 partners has been obtained. All the other pictures have been generated by Onera in the framework of AG2 project.
Rev 2 and Rev3 asked some details of the wing design. As the paper is related to morphing technology, I made a summary of the wing design in the first version. New chapters describing the wing design and the numerical methods used have been added.
Rev2 asked to add the flight conditions for each figure caption: done. “Sea level” has been replaced by “Altitude=0ft”.
Rev 2: explanation of “Delta” on last figures. It was the deflection angle of the last tab segment. For homogeneity with flap system, it has been replaced by “Delta3” and “Tab Deflection” has been added in the figures.
Some words about CRIAQ works have been added, as suggested by Rev 3.
Rev 3: what about testing in wind tunnel ?
There is a follow on project considering wind tunnel tests (GRETEL) about technologies investigated in AG2 (not only High-lift systems, but also new devices for load control). The “problem” is that the planform selected is not the AG2-NLF aircraft, but the reference aircraft design in Clean Sky 1. There were different arguments for that, the main one being that a complete aircraft model is available. The project leader considered that it was possible to adapt the different techno evaluated to the original wing. In my opinion, this is right except for the flap system that has to be redesigned. And taking into account the fact that ONERA is not involved in GRETEL project (except for some tests), I
didn’t make some comments in my paper. However, maybe contributors to other chapters of the Special Issue, dealing with load control, could speak about it.
Rev 1: Apparently, it was not clear if the paper was a review paper of a technical one. The paper is a technical one, but to introduce the work done in AG2, I made an introduction that may be a little bit too long … But I think it is necessary in order to introduce how morphing technique lead to aerodynamic improvements on this configuration, which is not the usual way we have in mind when considering morphing. In general, morphing technologies are investigated at high speed conditions. In this study, the use of morphed flap is mandatory if we want to respect the requirement to have no external fairings for flaps, and the system defined can be used in off design conditions to extend the flight domain.
As Rev2 and Rev3 considered the introductions as OK in their comments, I have not changed its structure, but I have added sub-headers for each applications. I hope it helps for the readers understanding.
Finally, the paper focus on aerodynamic performance improvement thanks to morphing technology. Not the details about how the morphed shape is obtained, except if it leads to a constraint for the design (for instance: how the multi functional flap system is segmented). I guess that all these details will be provided by other contributors to the special issue as there are dedicated chapters for system design, materials, structures etc … Therefore, all the suggested reference will be considered in these chapters.
MOENS Frédéric
ONERA
Reviewer 3 Report
SUBJECT: PAPER REVISION - Augmented Aircraft Performance by the use of Morphing Technology for a Turboprop Regional Aircraft Wing (REF.: MS. NO. biomimetics-548762)
The manuscript deals with an interesting investigation concerning the aerodynamic study of an integrated morphing system: both a variable-shape LE and a TE have been assessed with reference to a green regional aircraft. The architecture of the devices, already discussed in the literature, has been well summarized in order to introduce the reader to the understanding of these prototypes. The theoretical analysis and respective results are outlined in order to highlight the system potentialities, emphasising at the same time the drawbacks of the current wing geometries.
By my side, the paper faces a very interesting topic which is expected to contribute on the field of “Smart Structures” treated by Biomimetics journal: the technical content of the paper is worthy of publication. Anyway, the reviewer suggests the following comments, to be considered and addressed in order this paper is considered for publication.
ï‚· Add a list comprising all the symbols and abbreviations;
ï‚· Introduce more precise details on numerical simulations. What are the hypotheses behind the modelling? What kind of turbulence model, if used, was implemented? What are the boundary conditions? Which SW/tools have been used?
ï‚· The author cited the SARISTU project well; he could also introduce other industrial experiences such as the CRIAQ project, even focused on the development of adaptive platforms;
ï‚· What comments about testing in the wind tunnel? Are such experiences planned? It would be interesting to add a little paragraph about it.
Author Response

(The authors gave the same response as above.)

Round 2
Reviewer 1 Report
No more comments.